# Scaffolding of Mitogen-Activated Protein Kinase Signaling by β-Arrestins

**DOI:** 10.3390/ijms23021000

**Published:** 2022-01-17

**Authors:** Kiae Kim, Yeonjin Han, Longhan Duan, Ka Young Chung

**Affiliations:** School of Pharmacy, Sungkyunkwan University, 2066 Seoburo, Suwon 16419, Korea; kimkiae95@skku.edu (K.K.); bbabbibboo@g.skku.edu (Y.H.); noraline@g.skku.edu (L.D.)

**Keywords:** arrestin, MAPK, scaffold, protein structure

## Abstract

β-arrestins were initially identified to desensitize and internalize G-protein-coupled receptors (GPCRs). Receptor-bound β-arrestins also initiate a second wave of signaling by scaffolding mitogen-activated protein kinase (MAPK) signaling components, MAPK kinase kinase, MAPK kinase, and MAPK. In particular, β-arrestins facilitate ERK1/2 or JNK3 activation by scaffolding signal cascade components such as ERK1/2-MEK1-cRaf or JNK3-MKK4/7-ASK1. Understanding the precise molecular and structural mechanisms of β-arrestin-mediated MAPK scaffolding assembly would deepen our understanding of GPCR-mediated MAPK activation and provide clues for the selective regulation of the MAPK signaling cascade for therapeutic purposes. Over the last decade, numerous research groups have attempted to understand the molecular and structural mechanisms of β-arrestin-mediated MAPK scaffolding assembly. Although not providing the complete mechanism, these efforts suggest potential binding interfaces between β-arrestins and MAPK signaling components and the mechanism for MAPK signal amplification by β-arrestin-mediated scaffolding. This review summarizes recent developments of cellular and molecular works on the scaffolding mechanism of β-arrestin for MAPK signaling cascade.

## 1. Introduction

G-protein-coupled receptors (GPCRs) constitute the largest membrane receptor family that mediates various physiological and pathological functions [1]. The canonical downstream signaling molecule for GPCRs is the heterotrimeric G-protein, which regulates second messenger (e.g., cAMP and Ca^2+^) generation [2]. Arrestins were first discovered to desensitize GPCR-mediated G-protein signaling [3]. A total of four arrestins (arrestin-1–4) have been identified in humans; arrestin-1 and -4 (visual and cone arrestins, respectively) are expressed exclusively in the visual system, whereas arrestin-2 and -3 (also known as β-arrestin 1 and 2, respectively) are ubiquitously expressed [3]. GPCR kinases (GRKs) phosphorylate agonist-activated GPCRs at their C-tail or intracellular loops, which provide binding sites for arrestins (Figure 1A). The arrestin binding sites on receptors are similar to the G-protein binding sites (Figure 1A,B); therefore, arrestin binding precludes receptor coupling to G-proteins, resulting in the desensitization of G-protein signaling [3,4]. However, recently, it was found that the GPCR–G-protein–β-arrestin megacomplex can exist at the endosome where arrestin does not bind to the receptor cytosolic core but interacts only at the phosphorylated C-tail [5,6].

Arrestin binding also induces clathrin-mediated endocytosis of receptors [3,4]. In this process, arrestin acts as a scaffolding protein, linking the receptor with internalization machinery molecules such as clathrin and AP-2 [4,7,8,9,10]. Likewise, arrestins have been reported to interact with numerous other proteins including signaling proteins, adapter proteins, and transcriptional regulators [11,12,13]. Interaction with arrestins may alter the cellular localization or activation status of the binding partners [11,12]. The interaction occurs either with the basal or GPCR-induced active state of arrestins [4,10,14]. Among these binding partners, the mitogen-activated protein kinase (MAPK) family is a well-characterized binding partner of arrestins. 

MAPKs are the most important signaling protein families that respond to diverse external stresses such as heat, mitogens, and cytokines [15]. MAPKs regulate various cellular functions such as gene expression, cell survival, differentiation, proliferation, and apoptosis [15]. MAPK activation occurs through cascades of phosphorylation by three components: MAPK kinase kinase (MAP3K), MAPK kinase (MAP2K), and MAPK. MAP3K is an upstream kinase for MAP2K; MAP2K is an upstream kinase for MAPK; and MAPK phosphorylates the downstream effector proteins to regulate their active status [15,16]. 

In mammals, there are four distinct MAPK groups: ERKs (ERK1 and 2), JNKs (JNK1, 2, and 3), p38 (p38α–δ), and ERK5, each of which has its own upstream MAP2Ks and MAP3Ks [16,17]. How the MAPK signaling components accomplish signaling fidelity has been a long-standing mystery. Many studies have suggested that the scaffolding proteins co-localize MAPK signaling components which either facilitate the signaling cascade or modulate the negative feedback of a specific MAPK signaling pathway [18,19]. MAPK scaffolding proteins also regulate the subcellular localization of MAPK signaling components. Thus, MAPK scaffolding proteins organize specific MAPK signaling components to link the input to proper biological outcomes [18,19,20,21,22]. Therefore, MAPK scaffolding proteins have been considered as potential therapeutic targets [20,23,24,25].

Several MAPK scaffolding proteins have been identified in mammals including JIP, KSR, MP1, IQGAP1, paxillin, MORG1, JSAP1, DUSP22, and arrestins [21,22,26,27]. Depending on the scaffolding proteins, their functional consequences and molecular mechanisms are different [18]. For example, KSR recruits Raf, MEK1, and ERK1/2 at the plasma membrane promoting the ERK1/2 activation while PEB (phosphatidyl-ethanolamine-binding protein 1/RKIP) binds to Raf and MEK1 preventing their physical interaction and MEK1 activation by Raf [22]. Therefore, to precisely understand and regulate a specific MAPK signaling pathway, it is important to study the detailed scaffolding mechanism of each MAPK scaffolding protein. In this review, we discuss the functional consequences of arrestin scaffolding of MAPK signaling components and the structural mechanism of scaffolding. 

## 2. Scaffolding ERK1/2 Signaling by β-Arrestin

Many previous studies have reported that GPCR activation induces ERK1/2 phosphorylation, which is either G-protein-dependent or -independent [28,29,30]. G_i_ or G_q_ activation has been shown to activate the ERK1/2 signaling cascades with unknown precise mechanisms. G-protein-mediated ERK1/2 activation occurs quickly after GPCR stimulation with a transient nature [28,29,30]. The G-protein-activated ERK1/2 tends to move into the nucleus where it phosphorylates downstream nuclear effector proteins [28]. 

G-protein-independent ERK1/2 activation after GPCR stimulation was first suggested by Lefkowitz group in 1999 [31]; β-arrestin was shown to be involved in c-Src recruitment to the receptor to activate ERK1/2. In this report, the authors claimed that β-arrestin initiates a second wave of receptor signal transduction. Soon after, the same research group published a paper showing that β-arrestin acts as a “scaffold” for cRaf, MEK1, and ERK1/2 to facilitate ERK1/2 activation [32]. ERK1/2 activation mediated by β-arrestin is distinct from that mediated by G-protein. β-arrestin-mediated ERK1/2 activation tends to occur slowly after GPCR stimulation in a sustained manner; moreover, the activated ERK1/2 either stays in the cytosol or moves to the nucleus depending on the stimulated receptor [28,29,30,33,34,35,36]. All these studies agree that β-arrestin activation by GPCRs is essential for ERK1/2 activation [29,32,33,37,38,39] and that the molecular mechanism of β-arrestin-mediated ERK1/2 activation is via β-arrestin’s scaffolding action of cRaf, MEK1, and ERK1/2 [32,35]. 

## 3. Structural Mechanism of Scaffolding ERK1/2 Signaling by β-Arrestin

The molecular mechanism of β-arrestin-mediated scaffolding of the ERK1/2 signaling components is a long-standing mystery. Several studies have suggested the structural mechanism of scaffolding using biochemical and biophysical tools such as mutagenesis, peptide array, molecular simulation, and hydrogen/deuterium exchange mass spectrometry (HDX-MS) [40,41,42,43,44]. Although not completely understood, these studies provide insight into how β-arrestins scaffold ERK1/2 signaling components.

Arrestins are composed of N- and C-domains (Figure 2A), which are connected by two linkers with the second linker bringing the C-terminal tail, including the β-strand XX (Figure 2A, orange), to the N-domain. The agonist-activated phosphorylated receptor binding induces conformational changes in arrestin (Figure 2B). The phosphorylated C-tail of the receptor interacts at the N-domain of arrestin, which displaces the β-strand XX; the loops between the two domains (finger loop, middle loop, C-loop, and lariat loop) undergo conformational changes; the activation also induces disruption of the ionic interaction between the two domains (i.e., polar core) and interdomain rotation [45,46,47,48,49,50]. Thus, the removal of the C-terminal tail or a disruption of the polar core could mimic the active conformation of arrestins [51,52]. As receptor-induced activation of β-arrestin is suggested to be required for ERK1/2 activation, the scaffolding mechanism of β-arrestin for ERK1/2 activation has been studied using these active state-mimicking constructs. 

The direct interaction between β-arrestin and cRaf was suggested by an early study that proposed β-arrestin as a scaffolding protein for ERK1/2 signaling components [32]. A mutation and co-immunoprecipitation study suggested the cRaf binding site on β-arrestin, in which the β-arrestin 1 R307A mutant failed to interact with cRaf [42]. Later, a molecular simulation study suggested that a region near R307 (i.e., back loop) of β-arrestin 1 interacts with a region near K84 of cRaf [43]. cRaf is composed of a Ras-binding domain (RBD), C1 domain (CRD), hinge region, and kinase domain (Figure 2C, left) [53,54], and K84 is located within the RBD. A recent hydrogen/deuterium exchange mass spectrometry (HDX-MS) study with purified β-arrestin 1 and cRaf RBD confirmed that the back loop of β-arrestin 1 (i.e., near R307) interacts with cRaf RBD (Figure 2D) [44]. It is interesting to note that the kinase activity of the Raf family is auto-inhibited by the N-terminal domains, including RBD and CRD, and the binding of Ras to RBD releases the N-terminal domain, leading to exposure of the kinase domain (Figure 2C, right) [53]. The observation that β-arrestin 1 interacts with the RBD of cRaf prompted the hypothesis that β-arrestin 1 binding would release the N-terminal domains from the kinase domain, resulting in kinase domain activation (Figure 2D). This hypothesis was confirmed by a recent study in which the kinase activity of cRaf increased upon β-arrestin 1 binding [55]. 

Whether β-arrestin 1 needs to be activated by a phosphorylated receptor for cRaf binding remains controversial. Our results, with HDX-MS and in cell BRET assays, suggest that activated receptors are not required for β-arrestin 1 and cRaf interaction, while a recent study using a GST pull-down assay showed an increased β-arrestin 1 and cRaf interaction when V2Rpp, a phosphorylated C-terminal peptide of the V2 vasopressin receptor, was co-incubated [44,55]. Therefore, further studies are needed to fully understand whether receptor-mediated β-arrestin 1 activation is required for cRaf binding. Moreover, the involvement of β-arrestin’s scaffolding action for A-Raf or B-Raf needs to be investigated.

The binding interface between β-arrestin and MEK1 has been suggested in a few studies, and these studies have shown different results [40,41,43,44]. Meng et al. suggested that residues 46–70 in the N-terminal region of MEK1 and D26 and D29 in the N-domain of β-arrestin 1 are critical for the interaction [40], and a molecular simulation study suggested that D29 and E35 of β-arrestin 1 interact with a region near E272 of MEK1 [43]. However, our recent study using HDX-MS and fluorescence quenching analysis suggested that the N-lobe of MEK1 interacts with the interdomain loop I of β-arrestin 1 (Figure 2D) [56]. The discrepancies between different studies may be due to the different experimental techniques that were used; Meng et al. used peptide arrays, while our HDX-MS study used whole proteins to define the binding interfaces. Nevertheless, the requirement of GPCR-mediated β-arrestin activation for MEK1 binding is consistent between the previous studies (Figure 2D) [43,44,57].

The interaction between β-arrestin and ERK1/2 is also increased by receptor-mediated β-arrestin activation [43,44,57]. Our recent HDX-MS and fluorescence quenching analysis study suggested that activated β-arrestin interacts with ERK1/2 between the gate loop of β-arrestin 1 and the N-lobe of ERK1/2 and between the interdomain loop II of β-arrestin 1 and the C-lobe of ERK1/2 (Figure 2D) [44]. The interaction at the interdomain loop II of β-arrestin was also suggested by Cassier et al. who observed that phosphorylation of threonine 383 on the interdomain loop II of β-arrestin 2 induced a conformational change in this region, which facilitated phosphorylated receptor binding and subsequent ERK interaction [37]. The interaction at the gate loop of β-arrestin was also proposed in previous studies; Xu et al. suggested that ERK2 interacts at the lariat loop through the gate loop of β-arrestin 2 [58], and Bourquard et al. predicted that the gate loop of β-arrestin 1 interacts with the N-lobe of ERK1 [43]. 

Whether the interaction of the three ERK1/2 signaling components with β-arrestin is cooperative has been comprehensively studied. On the one hand, it was previously reported that the cRaf interaction with β-arrestin 2 is not affected by simultaneous expression of either MEK1 or ERK2 and that the simultaneous expression of MEK1 and ERK2 does not affect the interaction of either kinase with β-arrestin 2 [32]. On the other hand, in the same study, the binding of MEK1 or ERK2 to β-arrestin 2 was enhanced by the co-expression of cRaf [32], and the authors suggested that the binding of cRaf to β-arrestin 2 facilitates the assembly of other ERK1/2 signaling components on β-arrestin. Further studies are needed to understand how cRaf binding facilitates MEK1 or ERK2 interactions to β-arrestin.

In summary, these studies propose that the assembly of the ERK1/2 signaling components on β-arrestin locates each upstream kinase to phosphorylate the downstream kinase (Figure 2D). Our recent study also suggested that cRaf and MEK1 bind to β-arrestin when the two kinases are in an active state. However, ERK1/2 may dissociate from β-arrestin when it is activated so that phosphorylated active ERK1/2 can leave the β-arrestin-mediated scaffolding assembly to phosphorylate downstream effector proteins and free up the space for the recruitment of inactive ERK1/2 into the scaffolding assembly [44]. 

## 4. Scaffolding JNK3 Signaling by β-Arrestin

The first clue for the scaffolding role of β-arrestin in the JNK signaling was also suggested by the Lefkowitz group in 2000 [59]; one year after the β-arrestin-mediated scaffolding of ERK1/2 signaling was suggested by the same research group [31]. In this study, the interaction between β-arrestin 2 and JNK3 was identified by a yeast-2-hybrid screening study, and the result was confirmed by co-immunoprecipitation. In the same study, β-arrestin 2 was co-immunoprecipitated with ASK1 and MKK4, the MAP3K and MAP2K of JNK3, respectively. Later, the direct interactions between β-arrestin 2 and JNK3 [41,56,60], β-arrestin 2 and MKK4 or MKK7 [61,62,63], and β-arrestin 2 and ASK1 [36,51] were confirmed by various experimental systems, suggesting that the assembly of three JNK3 signaling components on β-arrestin 2 facilitates the kinase cascade. Interestingly, most studies have shown that β-arrestin 2, but not β-arrestin 1, is the major subtype of the JNK3 signaling component scaffolding and activation. The β-arrestin-mediated scaffolding and activation of JNK1 or JNK2 is controversial; in the early study, β-arrestin failed to facilitate JNK1 activation [59] while in another study β-arrestin 2 facilitated the activation of JNK1 or JNK2 via scaffolding them [64]. 

Similar to the ERK1/2 scaffolding, the interaction between JNK3 and β-arrestin 2 results in cytosolic localization of JNK3 [59,60]. Unlike ERK1/2, however, whether receptor-mediated β-arrestin activation is required for the interaction between β-arrestin and JNK3 remains controversial. Early studies have suggested that receptor-mediated β-arrestin activation is not required [41,59,65] while more recent studies suggest that β-arrestin activation either by a receptor or other molecule, such as IP_6_, facilitates β-arrestin 2 and JNK3 interaction [63,66,67]. 

## 5. Structural Mechanism of Scaffolding JNK3 Signaling by Arrestins

Several studies have suggested binding interfaces between β-arrestin and JNK3 [41,56,59,63,67,68,69,70]. Earlier studies with mutation or domain swapping proposed that the C-terminal half of β-arrestin 2 is the binding site for JNK3 [59,68,71]. However, recent studies suggest that the first 25 residues of β-arrestin 2 are the key binding site for JNK3 (Figure 3A), although it is possible that other parts of β-arrestin 2 are also involved in JNK3 binding [56,63,67,70]. In the basal state, this region is blocked by the β-strand XX (Figure 2A and Figure 3A); therefore, the β-strand XX should be released for JNK3 interaction. This is consistent with previous reports in which a receptor or IP_6_-mediated β-arrestin 2 activation facilitates JNK3 interaction [63,66,67].

JNK3 is activated by phosphorylation at two residues, Thr221 and Tyr223, which are phosphorylated by MMK7 and MKK4, respectively. The binding interface between β-arrestin 2 and MKK4 or NKK7 has been less extensively studied than that between β-arrestin 2 and JNK3. Nevertheless, a few studies have suggested that MKK4 and MKK7 compete for the same binding sites on β-arrestin 2 [56,62]. A recent study by the Gurevich group observed that the first 25 residues of β-arrestin 2, the same region where JNK3 interacts, is the binding site for MKK4 and NKK7 (Figure 3A) [56]. In this study, the interaction between β-arrestin 2 and MKK4/7 increased when MKK4/7 was occupied with ATP, and the interaction between β-arrestin 2 and MKK7 was stronger than that between β-arrestin 2 and MKK4. Interestingly, the phosphorylation-induced activation of MKK4 and MKK7 resulted in a change in the binding site within the first 25 residues of β-arrestin 2. The same study also showed that the binding affinity between JNK3 and β-arrestin 2 was reduced when JNK3 was activated so that the active JNK3 was released from β-arrestin 2. Taken together, the authors suggested a “conveyor belt” model for the JNK3 signal amplification by β-arrestin 2 (Figure 3B); ATP-bound MKK4 and MKK7 compete for the similar binding site on β-arrestin 2 to phosphorylate JNK3; in turn, the dual-phosphorylated JNK3 leaves the β-arrestin-mediated scaffolding assembly so that unphosphorylated inactive JNK3 can bind to the assembly. 

The interaction mechanism between β-arrestin 2 and ASK1 has been reported in a few studies [41,56,69]. These studies found that the C-terminal half, including the kinase domain of ASK1, is the binding site for the first 25 residues of β-arrestin 2 (Figure 3A). As all four components of JNK3 activation (i.e., ASK1, MKK4, MKK7, and JNK3) interact at a similar region on β-arrestin 2, more studies are needed to precisely identify the binding interfaces.

## 6. Phosphorylation Barcode and MAPK Scaffolding by β-Arrestin

Different receptors have different phosphorylation patterns, and, even for the same receptor, they can have distinct phosphorylation patterns depending on the kinases. β-arrestins have several of phosphate-binding sites (Figure 4A) [72,73], and it has been suggested that binding of differently phosphorylated receptors induces different active conformations of β-arrestins, resulting in distinct downstream signaling pathways [46,47,72,74,75,76]. This sophisticated regulation of β-arrestin conformation and function by the phosphorylation pattern of the receptor is named the “barcode theory” (Figure 4B) [74].

Long before the barcode theory was suggested, it was reported that β-arrestin-mediated ERK1/2 activation is dependent on the GRK isoforms that phosphorylate the receptor [29,77]. GRK2 and GRK3 phosphorylate the V2 vasopressin receptor and β_2_-adrenergic receptor to be desensitized by β-arrestin recruitment. However, GRK5 and GRK6 phosphorylate receptors to induce β-arrestin-mediated ERK1/2 activation. Moreover, different ligands induced different receptor conformations and different phosphorylation patterns, resulting in different β-arrestin activation [78,79]. The phosphorylation sites on the receptors by different kinases have been identified [74], and more detailed studies with mutation studies, NMR, or X-ray crystallography have deepened our understanding of the correlation between the phosphorylation pattern of the receptor, β-arrestin conformation, and ERK1/2 activation [47,76,80,81]. For example, carefully designed phospho-peptides or phosphorylation site-deleting mutants showed distinct ERK1/2 activation properties [76,81]. However, a specific phosphorylation pattern for JNK3 activation has not yet been studied. Moreover, as there are no high-resolution structures of β-arrestin-MAPK, MAP2K, or MAP3K mega-complexes, the precise structural mechanism of biased downstream signaling by β-arrestin needs to be elucidated.

## 7. Summary

In this review, we discussed the molecular and structural mechanisms of β-arrestin-mediated scaffolding of ERK1/2 and JNK3 signaling components. In both cases, β-arrestin must be activated to complete the scaffolding assembly. Furthermore, it has been suggested that, upon activation, both ERK1/2 and JNK3, the most downstream kinase of MAPK signaling cascade, leave the β-arrestin-mediated scaffolding assembly so that inactive MAPK can be recruited, which results in signal amplification. 

However, the binding interfaces within the β-arrestin-mediated scaffolding assemblies seem to differ; cRaf, MEK1, and ERK1/2 have distinct binding interfaces to interact with β-arrestin, while ASK1, MKK4/7, and JNK3 share a similar binding interface to interact with β-arrestin. The binding interfaces with β-arrestin are different between cRaf and ASK1, MEK1 and MKK4/7, and ERK1/2 and JNK3. Thus, defining the precise interacting residues in the different β-arrestin-mediated scaffolding assemblies could pave the way for the selective regulation of arrestin-mediated ERK or JNK signaling. 

## Figures and Tables

**Figure 1 ijms-23-01000-f001:**
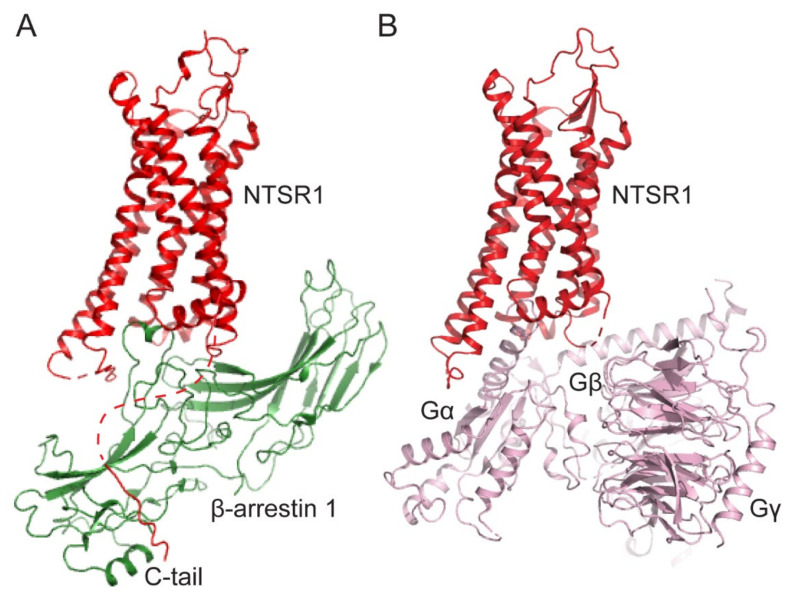
Comparison of GPCR-arrestin and GPCR-G-protein complexes. (**A**) The structure of the neurotensin receptor-1 (NTSR1, red) and β-arrestin-1 (green) complex (PDB: 6UP7). (**B**) The structure of the NTSR1 (red) and heterotrimeric G_i1_(light pink) complex (PDB: 7L0Q).

**Figure 2 ijms-23-01000-f002:**
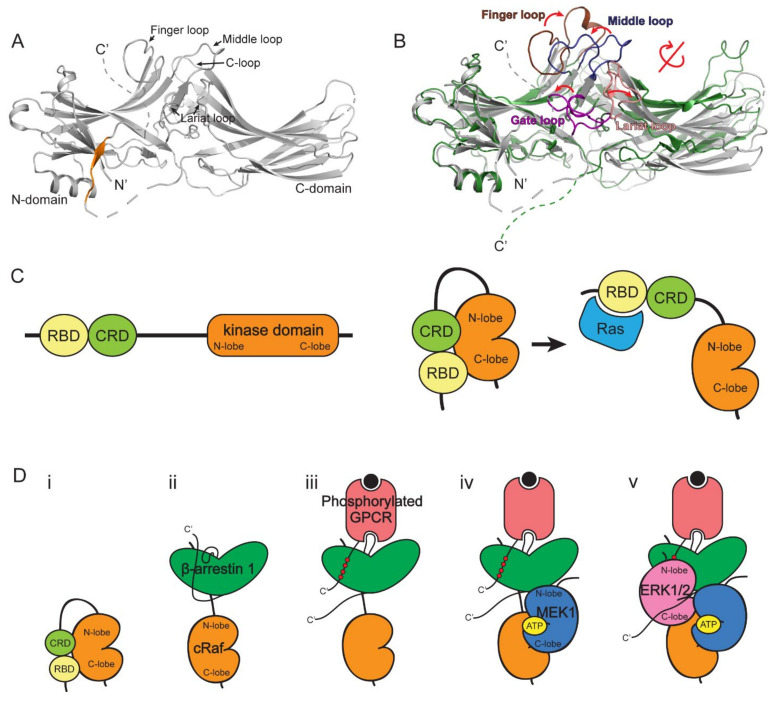
The structures of β-arrestin and the scaffolding mechanism of ERK1/2 signaling cascade. (**A**) The structure of β-arrestin 1 in the basal state (PDB: 1G4R). The C-terminal βXX strand is colored in orange. (**B**) Comparison of β-arrestin in the basal and active states. The structures of β-arrestin 1 in basal (grey, PDB: 1G4R) and NTSR1-bound active (green, PDB: 6UP7) states are compared. The loops on β-arrestin 1 are colored as follows: finger loop, brown; middle loop, dark blue; gate loop, purple; lariat loop, salmon. (**C**) The representation of cRaf domains (left) and conformational changes of cRaf upon Ras binding (right). (**D**) The proposed model of the scaffolding mechanism of ERK1/2 signaling cascade by β-arrestin 1. (i) cRaf is auto-inhibited by the N-terminal domains, including RBD and CRD. (ii) cRaf RBD interacts with the back loop of β-arrestin 1. (iii) The phosphorylated GPCR may bind to β-arrestin 1. (iv) The N-lobe of MEK1 interacts with the interdomain loop I of β-arrestin 1. (v) The activated β-arrestin 1 interacts with ERK1/2 between the gate loop of β-arrestin 1 and the N-lobe of ERK1/2 and the interdomain loop II of β-arrestin 1 and the C-lobe of ERK1/2.

**Figure 3 ijms-23-01000-f003:**
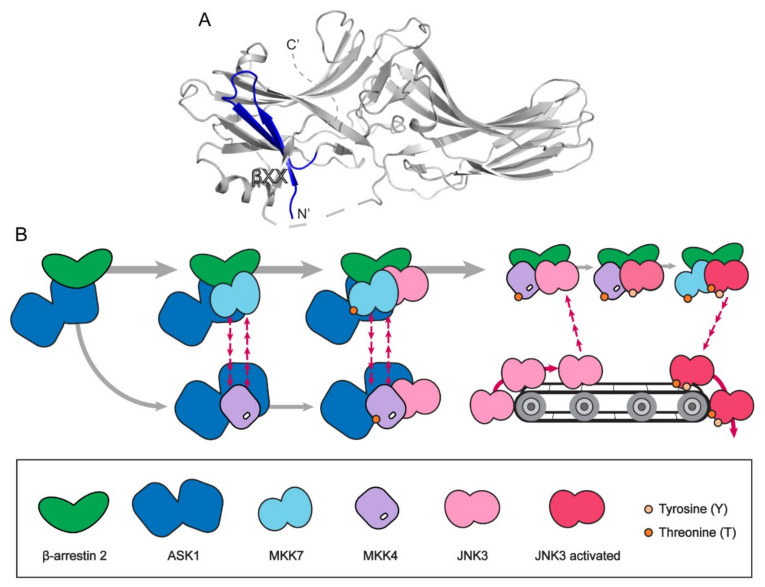
The JNK3 signaling scaffolding mechanism of β-arrestin 2. (**A**) The β-arrestin 2 structure with first 25 residues highlighted with blue (PDB: 3P2D). (**B**) The conveyor belt model of JNK3 signaling scaffolding mechanism of β-arrestin 2.

**Figure 4 ijms-23-01000-f004:**
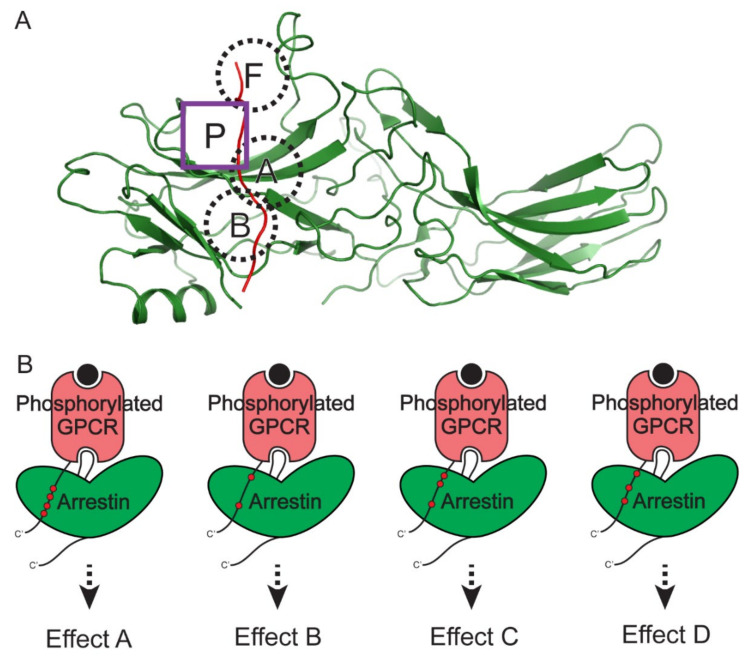
Distinct downstream signaling effects that are mediated by different phosphorylation patterns of GPCRs. (**A**) The four known phosphate-binding pockets on arrestins. The structure of CXCR7 phosphopeptide (red) and the activated β-arrestin 2 (green) complex (PDB: 6K3F) is shown as a representative structure. The conserved positively charged pockets on arrestins (pocket A, B, and F) are shown in the dashed black circles. The newly identified phosphate-binding pocket P on β-arrestin 2 is shown in the purple square. (**B**) Differently phosphorylated GPCRs induce differently activated β-arrestins, leading to distinct downstream signaling effects.

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
