# Peer review of "Scaffolding of Mitogen-Activated Protein Kinase Signaling by β-Arrestins"

_ijms, 2022, doi:10.3390/ijms23021000_

Round 1

Reviewer 1 Report

G protein coupled receptors (GPCRs) are the largest group of membranal proteins that mediate cellular responses to a wide variety of extracellular agents. GPCRs function via heterotrimeric G proteins, as well as G-protein independent mechanisms that transmit signals to signaling pathways. Although most of the activities of the GPCRs are mediated by second messengers,  some of their downstream signaling pathways were shown to act via the adaptor proteins arrestins. These proteins bind to the c-terminus of the GPCRs upon their stimulation and in many cases induce their desensitization. However, it was found β-arrestin  can also activate directly components of the MAPK cascades including JNK and ERK. These is mediated by a direct interaction of β-arrestin  with these MAPK components. In this review, the authors describethe functional consequences of arrestin scaffolding of MAPK signaling components and the structural mechanism of scaffolding. Overall, it is a well-written review, that provides most of the important information in the field. I have only minor comments that should be addressed prior to publication as follows:

1) Schemes like the one shown in Fig. 2 have already been published in many previous reviews, and therefore I am not convinced that this is at all necessary here. However, if the authors believe that it is essential, they should make it somewhat more informative and accurate. For example, addition of MAP4Ks and MAPKAPKs, mentioning more components in the MAP3K levels, the gene names (e.g. MAPK1 is ERK2, etc...) and the main effects of each of the cascades. In addition, it should be noted that the term SAPK is not really used anymore, but if the authors want to revive it, than it should appear probably for p38 as well.

2) In continuation with point 1, it seems to me that most of the references used for the MAPK cascades are outdated.  It is recommended to bring more current reviews in the field. This includes more information on the role of scaffolding in MAPK signaling.

3) In Fig. 3C, it is recommended to mention that the kinase is cRaf.  In addition, it would be interesting to speculate in the text whether A or B-Rafs may be involved as well.

4) Regarding JNKs, I agree that most of the available information is on JNK3-arrestin. However, there are a few papers that mention that JNK1 and JNK2 may be activated by arrestin as well. It is worthwhile mentioning it.

5) Can the authors provide examples for the distinct effects of the phosphorylation barcodes mentioned in Fig. 5 and the text describing it?

6) In the introduction the authors mention a GPCR–G protein–β-arrestin megacomplex at the endosomes that does not bind to receptors. Although interesting, this point is not further developed in the text, and I suggest to do it.

Author Response

Thank you for the reviewing our manuscript. Our responses to the reviewer’s comments are added as below.

1) Schemes like the one shown in Fig. 2 have already been published in many previous reviews, and therefore I am not convinced that this is at all necessary here. However, if the authors believe that it is essential, they should make it somewhat more informative and accurate. For example, addition of MAP4Ks and MAPKAPKs, mentioning more components in the MAP3K levels, the gene names (e.g. MAPK1 is ERK2, etc...) and the main effects of each of the cascades. In addition, it should be noted that the term SAPK is not really used anymore, but if the authors want to revive it, than it should appear probably for p38 as well.

Response: Thank you for pointing this out. We also agree that this figure is not necessary, and therefore we have removed this figure from the manuscript.

2) In continuation with point 1, it seems to me that most of the references used for the MAPK cascades are outdated. It is recommended to bring more current reviews in the field. This includes more information on the role of scaffolding in MAPK signaling.

Response: A few more recently published references are now added (DOI: 10.3389/fphar.2016.00037, 10.1371/journal.pone.0164259, 10.1080/14728222.2017.1311325, 10.3389/fphys.2012.00475).

3) In Fig. 3C, it is recommended to mention that the kinase is cRaf. In addition, it would be interesting to speculate in the text whether A or B-Rafs may be involved as well.

Response: It is described in the Fig. 3C legend that the kinase is cRaf. The involvement of arrestin for A or B-Raf scaffolding has not been studied yet. We have now added following sentence in the manuscript text.
“Moreover, the involvement of β-arrestin’s scaffolding action for A-Raf or B-Raf needs to be investigated.”

4) Regarding JNKs, I agree that most of the available information is on JNK3-arrestin. However, there are a few papers that mention that JNK1 and JNK2 may be activated by arrestin as well. It is worthwhile mentioning it.

Response: We now have described the interaction between arrestin and JNK1/2 in the manuscript text.
“The β-arrestin-mediated scaffolding and activation of JNK1 or JNK2 is controversial; in the early study, β-arrestin failed to facilitate JNK1 activation while in another study β-arrestin 2 facilitated the activation of JNK1 or JNK2 via scaffolding them.”

5) Can the authors provide examples for the distinct effects of the phosphorylation barcodes mentioned in Fig. 5 and the text describing it?

Response: The following sentence is added in the manuscript text.
“For example, carefully designed phospho-peptides or phosphorylation site-deleting mutants showed distinct ERK1/2 activation properties.

6) In the introduction the authors mention a GPCR–G protein–β-arrestin megacomplex at the endosomes that does not bind to receptors. Although interesting, this point is not further developed in the text, and I suggest to do it.

Response: The study about GPCR–G protein–β-arrestin megacomplex is just developing. Although the G protein-induced signaling in the megacomplex has been suggested, the signaling via β-arrestin in the megacomplex has not been studied. Therefore, we have not further discussed about this issue.

Reviewer 2 Report

The review by Kim et al. titled “Scaffolding of mitogen-activated protein kinase signaling by β-arrestins” is a well-written and comprehensive recount of recent findings on β-arrestins’ role in mediating MAPK signaling. The authors did a good job of discussing what future studies should focus on, without excessive self-promotion. This reviewer thinks that the review by Kim et al. warrants publication in the International Journal of Molecular Sciences upon some minor edits listed below.

1-  The authors should consider giving a couple of brief examples for the statement on page 3 lines 72-73:

“Depending on the scaffolding proteins, their functional consequences and molecular mechanisms are different[18]”

2- In the legend for Figure 3, the authors should  briefly explain (verbally) the molecular events that are taking place in steps “i” through “v” of Figure 3D.

3- In lines 197-199 of page 6, the author state that the regulation of JNK3 seems to be specific to β-arrestin 2 as opposed to β-arrestin 1. The review would benefit from molecular/structural/domain differences between different arrestins which might account for the ERK vs JNK preference for different arrestins.

4- Just in the case of JNK3, MKK4/7 and ASK1 described in lines 192-194 and Figure 2, is there any study that shows all three MAP kinases cRaf, MEK, and ERK1/2 to be bound to arrestin at the same time in concordance with its scaffolding role? Figure 3D makes it look like all three MAPKs can bind arrestin-GPCR complex at the same time, but I missed in the text any description of a co-precipitation or pulldown experiment showing the data for it.

5- Briefly re-state the outstanding questions and future directions the field should focus on in the Summary section.

Reviewer 3 Report

The manuscript by K. Kim, et al reviews intertwining between the beta-arrestin and the MAPK signalling. In general, the manuscript is well written. Still, there are some aspects that could be addressed to improve the readability of the paper:

  1. Formatting issues:
    1. The figures should not have word ‘Figure’ in the upper part of the graphics.
    2. Line 80: in Gi and Gq, the letter I and q should be typed as subscripts.
    3. Figure 3: in several parts, the colour code could be improved. For instance, orange could be used instead of yellow in case of betaXX strand (part A), and the label can be omitted from the graphics because the legend is sufficiently detailed. The labels of different domains and loops could be recoloured (dark blue?) especially in part B where the text overlays with the graphics. The light green and grey colours of overlayed cartoons in part B are visually hard to distinguish.
    4. Figure 5, part A: in the graphics, the pockets are denoted as A-B-F, whereas the legend text says A-B-C. The letter denoting the pocket P in the graphics could also be differently coloured (not just the circle dashes).
    5. There are some problems with the DOI-number spacing in references 14-16, 18-19, and 22-23.
  2. I would ask the authors to comment on the following aspects, whether as answers to reviewer, or within the manuscript text:
    1. Most of the reviewed experimental studies (especially the older ones, from 1990’s) were carried out utilizing high concentrations of recombinant proteins in vitro or over-expression of the tagged protein constructs in cellulo. Can the authors provide any data regarding the native expression levels of the MAPK cascade components and beta-arrestins in cells, and speculate on the possibility of artefacts originating from the use of elevated protein concentrations in the experimental systems?
    2. Is there any data regarding the effect of beta-arrestin interaction with MAPK cascade components on the efficiency/kinetics of the inhibitors of MAP3K, MAP2K, or MAPK? I.e., can the inhibition potency of the compounds depend on the scaffolding status of the MAP kinases?
    3. The authors provide no quantitative data regarding the strength of interactions (e.g., KD value) with the beta-arrestin:MAPK complexes, although in lines 223-235, there is a slight hint that such parameters have been assessed (increased/stronger interaction) – maybe the authors could expand their statement with some quantitative characteristic. Furthermore, is it likely that such scaffolding also affects the catalytic properties of the kinases, or does it mostly affect the intracellular localization of the latter?
  3. Further suggestions:
    1. It would be great to add an image of the kinome tree with marked kinases mentioned in the manuscript (not only MAP pathway, but also GRKs). The authors could even consider using a colour code for denoting different hierarchy levels in the MAPK cascade, or various MAPK groups. Alternatively, the authors could provide the BLAST-comparison of amino acid sequences between the ERK1/2, p38, JNK, and ERK5/BMK1. This would add a much-appreciated level of detail regarding the extent of differences between the various MAPK groups.
    2. From the abstract, it is difficult to decide for the reader whether the main text focusses rather on the cellular experiments, in vitro experiments with recombinant proteins, or both. The authors could add a couple of remarks on the reviewed experimental systems.

Author Response

Thank you for the reviewing our manuscript. Our responses to the reviewer’s comments are added as below.

  1. Formatting issues:
    1. The figures should not have word ‘Figure’ in the upper part of the graphics.
      Response: It is corrected.

    2. Line 80: in Gi and Gq, the letter I and q should be typed as subscripts.

Response: It is corrected.

    1. Figure 3: in several parts, the colour code could be improved. For instance, orange could be used instead of yellow in case of betaXX strand (part A), and the label can be omitted from the graphics because the legend is sufficiently detailed. The labels of different domains and loops could be recoloured (dark blue?) especially in part B where the text overlays with the graphics. The light green and grey colours of overlayed cartoons in part B are visually hard to distinguish.
      Response: The figures are modified.

    2. Figure 5, part A: in the graphics, the pockets are denoted as A-B-F, whereas the legend text says A-B-C. The letter denoting the pocket P in the graphics could also be differently coloured (not just the circle dashes).
      Response: The figure is modified, and the legend is corrected.

    3. There are some problems with the DOI-number spacing in references 14-16, 18-19, and 22-23.
      Response: The errors are corrected.

  1. I would ask the authors to comment on the following aspects, whether as answers to reviewer, or within the manuscript text:
    1. Most of the reviewed experimental studies (especially the older ones, from 1990’s) were carried out utilizing high concentrations of recombinant proteins in vitro or over-expression of the tagged protein constructs in cellulo. Can the authors provide any data regarding the native expression levels of the MAPK cascade components and beta-arrestins in cells, and speculate on the possibility of artefacts originating from the use of elevated protein concentrations in the experimental systems?

      Response: The endogenous β-arrestin concentration was reported to be 80 nM (DOI: 10.3389/fendo.2017.00032). However, upon stimulation, β-arrestin translocate to specific regions of the cell such as plasma membrane or endosome, and the local concentration of β-arrestin could be higher than the average cellular concentration. Moreover, arrestin was measured as 2 µM in many cells such as HEK293. (Unpublished data, measured by arrestin antibody and using purified arrestin as control). We could not find the cellular concentration of MAPK signaling components.
      The in vitro system cannot fully mimic the physiological environment such as salt concentration, lipid membrane, charge distribution, and other factors, and therefore the affinity between two proteins is usually lower in vitro than in vivo. Thus, to form protein-protein complex in vitro, it is common to use higher concentration of proteins.

    2. Is there any data regarding the effect of beta-arrestin interaction with MAPK cascade components on the efficiency/kinetics of the inhibitors of MAP3K, MAP2K, or MAPK? I.e., can the inhibition potency of the compounds depend on the scaffolding status of the MAP kinases?

      Response: This is very intriguing question. Unfortunately, as far as we know, there is no data about the effect of beta-arrestin interaction on the activity for the MAPK cascade component inhibitors.

    3. The authors provide no quantitative data regarding the strength of interactions (e.g., KD value) with the beta-arrestin:MAPK complexes, although in lines 223-235, there is a slight hint that such parameters have been assessed (increased/stronger interaction) – maybe the authors could expand their statement with some quantitative characteristic. Furthermore, is it likely that such scaffolding also affects the catalytic properties of the kinases, or does it mostly affect the intracellular localization of the latter?

      Response:
      There is no study measuring the Kd values of the interaction between arrestin and MAPK signaling components. In our hands, when we analyzed HDX-MS, the HDX-MS profile change differences are detected, which could give us hint on the relative binding affinity but could not provide quantitative values.
      Most studies measured ERK1/2 or JNK3 activation (i.e., phosphorylation) to investigate whether arrestin-mediated scaffolding facilitates the signaling cascade. However, the effects of arrestin binding on the catalytic property itself also has not been studied with one exception, cRaf. The effects of arrestin binding on cRaf activity is already discussed in the manuscript text.
      “It is interesting to note that the kinase activity of the Raf family is auto-inhibited by the N-terminal domains, including RBD and CRD, and binding of Ras to RBD releases the N-terminal domain, leading to exposure of the kinase domain (Fig. 2C, right) [53]. The observation that β-arrestin 1 interacts with the RBD of cRaf prompted the hypothesis that β-arrestin 1 binding would release the N-terminal domains from the kinase domain, resulting in kinase domain activation (Fig. 2D). This hypothesis was confirmed by a recent study in which the kinase activity of cRaf increased upon β-arrestin 1 binding [55].”

  2. Further suggestions:
    1. It would be great to add an image of the kinome tree with marked kinases mentioned in the manuscript (not only MAP pathway, but also GRKs). The authors could even consider using a colour code for denoting different hierarchy levels in the MAPK cascade, or various MAPK groups. Alternatively, the authors could provide the BLAST-comparison of amino acid sequences between the ERK1/2, p38, JNK, and ERK5/BMK1. This would add a much-appreciated level of detail regarding the extent of differences between the various MAPK groups.

      Response: Thank you for the suggestion. It would be certainly interesting to show kinome and differences between different MAPKs. However, this is a review paper not an bio-informatics analysis paper. This type analysis would provide an insight on the currently discussed topic, and the authors would perform this type of analysis for our further study.

    2. From the abstract, it is difficult to decide for the reader whether the main text focusses rather on the cellular experiments, in vitro experiments with recombinant proteins, or both. The authors could add a couple of remarks on the reviewed experimental systems.

      Response: The following sentence is added in the manuscript text.
      “This review summarizes recent developments of cellular and molecular works on the scaffolding mechanism of β-arrestin for MAPK signaling cascade.”